# Lightweight Visual Transformers Outperform Convolutional Neural Networks for Gram-Stained Image Classification: An Empirical Study

**DOI:** 10.3390/biomedicines11051333

**Published:** 2023-04-30

**Authors:** Hee E. Kim, Mate E. Maros, Thomas Miethke, Maximilian Kittel, Fabian Siegel, Thomas Ganslandt

**Affiliations:** 1Department of Biomedical Informatics at the Center for Preventive Medicine and Digital Health (CPD), Medical Faculty Mannheim, Heidelberg University, Theodor-Kutzer-Ufer 1-3, 68167 Mannheim, Germany; 2Institute of Medical Microbiology and Hygiene, Medical Faculty Mannheim, Heidelberg University, Theodor-Kutzer-Ufer 1-3, 68167 Mannheim, Germany; 3Institute for Clinical Chemistry, Medical Faculty Mannheim, Heidelberg University, Theodor-Kutzer-Ufer 1-3, 68167 Mannheim, Germany; 4Chair of Medical Informatics, Friedrich-Alexander-Universität Erlangen-Nürnberg, 91054 Erlangen, Germany

**Keywords:** Gram-stain analysis, classification, deep learning, quantization, vision transformer, convolutional neural network

## Abstract

We aimed to automate Gram-stain analysis to speed up the detection of bacterial strains in patients suffering from infections. We performed comparative analyses of visual transformers (VT) using various configurations including model size (small vs. large), training epochs (1 vs. 100), and quantization schemes (tensor- or channel-wise) using float32 or int8 on publicly available (DIBaS, *n* = 660) and locally compiled (*n* = 8500) datasets. Six VT models (BEiT, DeiT, MobileViT, PoolFormer, Swin and ViT) were evaluated and compared to two convolutional neural networks (CNN), ResNet and ConvNeXT. The overall overview of performances including accuracy, inference time and model size was also visualized. Frames per second (FPS) of small models consistently surpassed their large counterparts by a factor of 1-2×. DeiT small was the fastest VT in int8 configuration (6.0 FPS). In conclusion, VTs consistently outperformed CNNs for Gram-stain classification in most settings even on smaller datasets.

## 1. Introduction

The progress of deep learning (DL) and artificial intelligence is astonishing, and it attracts numerous researchers and practitioners from multidisciplinary domains. Although tremendous literature regarding DL applications has been published in the medical domain [1], it is uncommon that DL applications are actually deployed in the clinical routine. In addition to common issues such as strict medical device regulations [2], interpretability and responsibility of DL models [3], researchers and practitioners also face multiple technical challenges in utilizing DL solutions, e.g., hardware capacities in hospitals are limited, and cloud computing or edge networks are also uncommon because of data privacy concerns [4]. Therefore, medical DL applications are often deployed to resource-limited devices, resulting in performance degradation.

Lightweight DL is an especially crucial subject when it comes to infectious diseases. According to Seymour et al. [5], in-hospital mortality could be lowered if antibiotics were administered within an initial three-hour window of sepsis care, which is remarkably time-sensitive. Gram-stain analysis is a rapid laboratory test that classifies bacterial species into two groups: either Gram-positive or Gram-negative [6]. It aims to shorten the time needed to correctly classify the underlying bacteria in sepsis patients and ultimately aims to decrease the time to a targeted treatment, which is the interval from symptom onset and diagnosis to the application of the therapy of the disease. Although a physician instantly prepares antibiotic therapy for a patient in practice, precise and rapid identification of an exact microorganism still matters for tailored treatment. This task currently relies on medical professionals [7] and it can be partially automated by DL solutions [8]. At this time, only a few studies utilize DL for Gram-stain analysis. Liu et al. [9] utilized six machine-learning algorithms and identified two species of Gram-positive bacteria, *B. megaterium* and *B. cereus,* by harnessing spectral features of Gram-stained images. Smith et al. [10] proposed a classification model by means of a convolutional neural networks (CNN) model, however, the solution took 9 min to classify a whole-slide image comprising 4.1 million pixels. Recently, our research group has demonstrated that by applying pruning and quantization model size (15×) and inference time (3-4×) of CNN can be substantially reduced and accelerated on limited edge devices such as smartphones without sacrificing accuracy [11]. However, visual transformer (VT) models, the state-of-the-art methodology in computer vision, have not yet been investigated.

CNN had been the de-facto DL architecture in the computer vision community since AlexNet [12] won the ImageNet Challenge in 2012. However, a marked paradigm shift occurred in 2020 when Google Brain Team introduced vision transformer (ViT) [13]. In fact, ViT is not a novel model architecture, but it has developed from the standard transformer encoder [14] from the natural language processing (NLP) domain. The performance of transformer models attained higher accuracy compared with the best-performing CNN model (e.g., ResNet [9]) on classification [15]. The mechanism for understanding images differs considerably between CNN and VTs. CNN captures a certain type of spatial structure present in the given dataset because they utilize spatial inductive biases that allow them to learn the local representations [16]. Inductive bias is a set of assumptions that can generalize a dataset and does not require large datasets compared with transformers-based models. On the other hand, VTs learn global representations by using self-attention mechanisms [16]. Multiple studies demonstrated that global representations triumph over inductive bias when trained on sufficiently large scales of datasets, as ViT surpassed ResNet with 300 M images [13].

This paper aimed to provide a guideline to researchers and practitioners on VT model selection as well as optimal model configuration for Gram-stained image classification. For this, six VT models were investigated using target metrics such as accuracy, inference time and model size and were benchmarked against two well-established CNN models. All models were compressed to 8-bit and were interoperable using the ONNX framework.

## 2. Background

A VT comprises three major components, as shown in Figure 1. (1) Linear projection takes input images and outputs joint embeddings. It splits images into predetermined-size patches that are flattened to linear patch embeddings added by positional embeddings. The transformer encoder takes these joint embedding vectors as input, also referred to as tokens, and returns the same length of weighted vectors as output. A class embedding is also attached to the input embeddings for the classification. The key element of the encoder is (2) Multi-head self-attention layers (MSA) and it takes three vectors, namely, query, key and value, while “*self*” indicates that query, key and value are identical. Attention is a weighted sum of value vectors and the weight is the inner product of a single query vector and a set of key vectors. Multi-head indicates that multiple attention modules process data in parallel. Finally, (3) Multilayer perceptron (MLP) is a fully connected neural network that classifies input images. The corresponding mathematical notation is found in the original paper [14].

Since the introduction and great success of the ViT model [13] by Dosovitskiy et al., numerous VT models and their applications have been proposed in the computer vision community. Despite VT models growing rapidly, they fall into one of five architecture categories and each architecture distinctively differs from one another. ViT is the (1) original VT model and its architecture is identical to the encoder block of the transformer model inherited from Natural Language Processing [14]. The authors demonstrated that ViT outperformed CNN, however, it required quadratic time complexity with respect to input image size and large data (300 million images) to pre-train. Therefore, many researchers have proposed innovative architectures to tackle the problem of the ViT model. (2) Multi-stage models introduced limited size of attention such as localized attention or sparse attention and processed feature vectors gradually and progressively. This mechanism was able to lighten the computational burden and resulted in linear computational complexity. Such an archetype model was the hierarchical vision transformer using shifted windows (Swin) [17] introduced by Liu et al. from Microsoft in 2021. Similarly, pyramid ViT (PVT) [18] and focal transformer models [19] are hierarchical VT models that introduce spatial reduction attention inspired by CNN’s backboned pyramid structure for dense prediction tasks [20,21]. More recently, Hassani and Shi proposed a hierarchical VT based on neighborhood attention that can capture a more global context [22]. (3) Knowledge distillation is another solution that is capable of training VT efficiently. Tourvron et al. from Facebook AI designed data-efficient image transformers (DeiT) that utilized distillation tokens to learn from a teacher agent. On the other hand, Ren et al. introduced a cross inductive bias distillation (CiT) [23] with an ensemble of multiple lightweight teachers instead of a single heavy and highly accurate teacher agent. Unlike conventional knowledge distillation models that are matching teacher to student in a one-to-one spatial relationship, Lin et al. proposed a one-to-all spatial matching knowledge distillation VT [24], which surpassed other models by a large margin. The (4) self-supervised model was inspired by BERT [25] and rooted in the NLP domain. It slices a given image into multiple patches referred to as “visual tokens’’ and randomly drops some patches. The model learns the generic features of images in an unsupervised manner by recovering the eliminated visual tokens. The generative pretraining from pixels (imageGPT) [26] is the same as GPT-2 [27] except for the activation and normalization layers. It outperformed a supervised model, ResNet. The drawback of imageGPT is the time complexity because its architecture learns images based on pixels instead of image patches. Bidirectional encoder representation from image transformers (BEiT) [28] is the most cited self-supervised model proposed by Bao et al. in Microsoft. It surpassed imageGPT by a large margin with much fewer parameters while concurrently outperforming two supervised VT models (ViT and DeiT). Finally, (5) hybrid type captures local and global representations by incorporating one or more components from CNN that could save on the computation burden by a large margin [29]. The idea of integrating inductive bias into global representations attracted numerous researchers. Multiple studies such as BoTNet [30], CMT [31], CvT [32], LeViT [33] and ViTc [34] improved accuracy and computational efficiency by combining convolutional layers to the VT model. MobileViT [35] was designed by Apple for efficient computation on mobile devices, however, it is more similar to CNN models than VTs. Furthermore, models such as PiT [36] and PoolFormer [37] achieved competitive results by incorporating pooling layers without attention layers or convolutional layers.

Based on their properties and the Gram-staining classification task at hand, we included the following VT models (BEiT, DeiT, MobileViT, PoolFormer, Swin and ViT) for systematic analyses and evaluation (Figure 2).

## 3. Materials and Methods

### 3.1. Data Set

Two Gram-stained image datasets were utilized in this study. One is the domestic dataset from Medical Faculty Mannheim, Heidelberg University (MHU) and the other is a publicly accessible dataset named DIBaS [38], the acronym for Digital Image of Bacterial Species. The MHU dataset consists of 8500 Gram-stained images collected from 2015 to 2019. The resolution of the images varied from 800 pixels by 600 pixels to 1920 pixels by 1080 pixels. In the given dataset, Gram-positive images (*n* = 5962) were two times more prevalent than Gram-negative images (*n* = 2766). On the other hand, the image size of DIBaS is identical to 1532-pixel by 2048-pixel. DIBaS contains only 660 images (20 images for 33 microorganisms) and it is also an unbalanced dataset where Gram-positive images (*n* = 280) and Gram-negative images (*n* = 194) are available. Therefore, an oversampling method [39] was applied to both of the datasets. For the MHU dataset, the number of Gram-negative images increased from 2766 to 5032 by applying rotation, while for the DIBaS dataset, we applied split and/or rotation to both classes and augmented Gram-positive images from 280 to 448 and negative images from 194 to 410. The augmented and balanced datasets were split into a training set (80%), a validation set (10%) and a test set (10%). Statistical evaluation methods such as cross-validation are uncommon among AI researchers because they are resource-intensive and time-consuming. Both datasets contain images cropped from whole slide images and contain one microorganism such as *Staphylococcus*, *Escherichia* or *Streptococcus*. The size of the images was rescaled to the same resolution as the pre-trained images (224 × 224 or 256 × 256) during the fine-tuning phase.

### 3.2. Study Design

We examined 128 models by accuracy, inference time and model size. The overview of the study design is shown in Figure 2. During the fine-tuning phase, 64 models were re-trained based on the combination of different models, epochs and datasets. Then, each model was compressed by two quantization strategies during the quantization phase. Briefly, eight models with minimum parameters and maximum parameters were fine-tuned to two custom Gram-stained image datasets with two epochs strategies, and then models were quantized either by channel or tensor.

The eight models included six VT models and two CNN models. Each model represents a distinctive architecture, which is summarized in Table 1. We chose the most cited model implementation among the same architectures. The two CNNs, ConvNeXT [40] and ResNet, served as baselines to be compared with VT models. ResNet was chosen because it is known to be a versatile and well-performing CNN architecture on various tasks [41], while ConvNeXT is a ResNet variation with hyperparameters that are similar to the ViT model. Furthermore, ConvNeXT outperformed the ViT model in a similar study classifying Gram-positive bacteria in a previous study [40].

All models were pre-trained on the ImageNet-1k dataset, which is a collection of 1.3 million images of subjects such as dogs and cats with 1000 classes. Note that each model can be used in various sizes (e.g., MobileViT-xxs, -xs and -s). They share the same architecture but differ in the number of model components (e.g., attention heads, encoder blocks, etc.). We examined each model with minimum (small) parameters and maximum (large) parameters. Furthermore, models were re-trained either for a single epoch or 100 epochs to examine the impact of the number of epochs on model accuracy during the fine-tuning phase. Two quantization strategies were applied to the models: (i) either the entire tensor (QT) as a whole or (ii) each channel separately (QC) was quantized from 32-bit float to 8-bit integer representation.

### 3.3. Metrics

The generalization capability of the models was evaluated by accuracy, which cares about the quantity of right or wrong decisions in unseen data. Accuracy is the most employed metric to measure the quality of a classifier, usually defined as true positives + true negatives divided by all samples. The F1-score [42] is often employed in conjunction with accuracy as a complementary metric for evaluating classifiers. Accuracy evaluates the quantity of right or wrong outcomes, whereas F1-score is a harmonic mean of precision and recall, which provides insight into whether a model is skewed to a certain class or not. F1-scores are reported in Section A.1.

### 3.4. Apparatus

To ensure reproducibility, all our analyses were performed in a containerized environment using a docker. The model tuning and evaluation were conducted in the following virtual environment: One NVIDIA Tesla V100 32 GB GPU was assigned to the docker container and one Intel Xeon Silver 4110 CPU and 189 GB of memory were shared from the host server. HuggingFace Optimum [43] v1.3.0 was utilized for re-training, model conversion and quantization.

## 4. Results

### 4.1. Fine-Tuning Progress

The history of the fine-tuning progress of all pre-trained models is visualized in Figure 3. The purple lines are the history of the models with minimum parameters referred to as “small model”, while the gray lines indicate models with maximum parameters referred to as “large model” respectively. Subplots in Figure 3a show that accuracy gradually increased over the learning cycle, especially the accuracy slope of MobileViT, ResNet and ViT, which rapidly gained accuracy compared with other models as the evaluation accuracies at the beginning of the epoch and the last stage of the epoch differ by a large degree on those three models. Moreover, the evaluation accuracy of BEiT and DeiT was depicted as relatively lower than other models during the fine-tuning phase, while ConvNeXT was the highest during the fine-tuning phase. With regard to the model size, the large models demonstrated higher accuracy compared with the small models, except for BEiT and ViT. All models encountered rapid overfitting when they were fine-tuned on the DIBaS data set (Figure 3b). In particular, ConvNext, DeiT and PoolFormer models immediately jumped to 100% validation accuracy regardless of the model size, while other models also attained 100% accuracy at the last epoch.

### 4.2. Accuracy and Quantization

The results of models re-trained for one epoch are illustrated in Figure 4. On the MHU dataset, the best accuracy was achieved by PoolFormer as follows: 93.2% for 1 epoch and 95.1% for 100 epochs. The highest accuracy on the DIBaS dataset was achieved by BEiT for 1 epoch (95.0%), respectively by ViT for 100 epochs (98.3%). Model accuracies were in the following range: BEiT (84.2–97.8%), ConvNeXT (49.4–92.8%), DeiT (80.6–92.3%), MobileViT (49.4–89.2%), PoolFormer (50.0–95.1%), ResNet (45.8–91.7%), Swin (49.4–93.2%) and ViT (85.7–98.3%). Overall, ViT showed the most well-rounded performance (always >85%) in these four settings (Figure 4a–d). Large BEiT and DeiT models suffered from performance degradation when undergoing channel-wise quantization (Figure 4d). Other models were sensitive to the dataset as they achieved competitive accuracy on the MHU dataset, but not on the DIBaS dataset. In particular, MobileViT large, PoolFormer small, ResNet and Swin small were sensitive to both dataset and epoch as they attained accuracy higher than 87.6% when they were re-trained for 100 epochs on the MHU dataset only.

### 4.3. Time, Size and Trade-Offs

We found no difference between model performances on the two datasets (MHU and DIBaS) in terms of inference time regardless of the model architecture and quantization approach (Figure 5). However, there were large differences mainly influenced by the model size. Frames per second (FPS) of small models consistently outperformed those of large models by a considerable margin, which was expected by design. The DeiT small model was able to process two times more images than the large model (5.9 images/s vs. 2.9 images/s). Models gained a minor improvement in FPS if they were quantized to integer8. BEiT, ConvNeXT, DeiT, PoolFormer and Swin accelerated 0.2–0.5 FPS, 0–0.5 FPS, 0.3–1 FPS, 0–1.2 FPS and 0–0.3 FPS, respectively. DeiT and ResNet small models were able to process at least five images per second (i.e., the results underlined in Figure 5), on the other hand, small BEiT and ViTs could process less than three images per second.

Next, we compared the overall evaluation of model size, accuracy and inference time visually using bubble charts (Figure 6). We notice that model accuracies on the MHU dataset outperformed compared to those on the DIBaS as the nodes consistently surpass 80% (y-axis, Figure 6a,b), whereas the position of the nodes varied from 50% to 98% accuracy (y-axis, Figure 6c,d). On the other hand, FPS was almost identical among similarly sized models, regardless of the datasets (x-axis, Figure 6). While the dispersion of FPS of small models (Figure 6a,c) was wider than that of the large models (Figure 6b,d). With regards to inference time, DeiT and ResNet classified more images than other models as they were consistently plotted on the upper-right quadrant of the plots.

## 5. Discussion

In this study, we performed a comprehensive comparison of six VT models and compared them to two CNN models. We examined their applicability to automated Gram-stained classification. Overall, VT models outperformed CNN models with fewer epochs and on a smaller dataset. Especially, VT models with ViT backbone (i.e., BEiT, Deit and ViT) were outstanding among other models. However, our findings have shown that model performances were determined not only by the model architecture but also by model configuration (e.g., epochs and quantization schemas) and the custom dataset. Hence, we advocate that the model architecture should be empirically determined by considering all of these parameters above.

With regard to the fine-tuning progress shown in Section 4.1, all models highly overfitted the DIBaS dataset. The high validation accuracy (Figure 3) did not guarantee high test accuracy (Figure 4) as five out of eight models (i.e., ConvNeXT, MobileViT, PoolFormer, ResNet and Swin) made a random guess on the DIBaS dataset for the testing phase (Figure 4c,d). We found that deep learning models suffer from the overfitting problem if the available data quantity is <1000 images. Regularization techniques (e.g. weight decay, weight normalization and batch normalization) have been previously proven to generalize models and address the overfitting problem. Weight decay [44] penalizes a large magnitude of coefficients, while batch normalization [45] rescales the layer’s input, and similarly, weight normalization [46] regulates the magnitude of learnable parameters. In addition to regularization techniques, early stopping [47] of the training process is also a widely applied strategy to avoid overfitting. It ends training if there is no improvement during the training-validation phase.

Both CNNs and VT classifiers achieved better results on a larger dataset (MHU) than on a smaller dataset (DIBaS). We found that BEiT, DeiT and ViT achieved high accuracies, regardless of the number of epochs and model size. This might be explained by their common backbone model: ViT [13]. We assume that global information learned by self-attention layers surpasses the value of learning local information by CNN. Architectures combining CNNs and VTs showed competitive accuracy results under certain conditions in this study. PoolFormer, which completely lacks an attention layer, showed the best accuracy on the MHU dataset when it was fine-tuned for 100 epochs, however, MobileViT, which consists of three VT blocks while having six CNN blocks and two additional convolutional layers, showed the lowest accuracy performances in average among VT models. BEiT and DeiT suffered from a considerable accuracy drop when they were quantized per channel (QC). These results are counterintuitive to the general belief [48] as QC is expected to obtain higher accuracies. QC provides a better and more sophisticated prediction because it consists of more parameters to train compared with QT models. It is possible that more intensive quantization made models overfit our custom dataset and failed to generalize. It might also explain the accuracy refinement from float32 models to int8 models although the improvement was marginal. We assume that removing or reducing the number of model parameters conveyed a similar impact as regularization techniques on a relatively small dataset.

Accuracy is a metric that captures the first impression of models, however, more insights could be gained when used with other metrics such as inference time and model size [49]. They are, in fact, non-trivial aspects of DL models in the context of deployment to resource-limited devices such as mobile devices (smartphones) without dedicated GPU resources. This is especially the case for patients suffering from an infectious disease because minimizing the time to diagnosis and the time to treatment is crucial for them [5]. This study demonstrated that the inference time in FPS units and the throughputs were enhanced on the models with smaller parameters and with the lower-bit presentation, as shown in Figure 5. These gains do not seem trivial, however, the optimization solutions can be scaled out when a model classifies a whole slide image (10k × 6.4k) which is equivalent to ~1.3k cropped images (224 × 224). An overview of all results is depicted in Figure 6 which might provide a guideline for a model selection on a selected dataset. As public Gram-stained data sets are extremely scarce (besides our local dataset (MHU), we found only one more additional public Gram-stained image dataset (DIBaS)), we could not perform systematic statistical comparisons. For this, most approaches require at least five (ideally) separate data sets to be able to infer non-parametric rank-based statistics [50,51].

Our study has other limitations. The scope of this study was limited to the image classification problem, although VT models have also made great progress on different problems such as object detection and segmentation [52]. Carion et al. from Facebook AI proposed DETR [53] for object detection which consists of both CNN and VT models. YOLOS [54] is another successful model for object detection inspired by DETR. SegFormer [55] is a hierarchical transformer encoder and a lightweight perceptron decoder for image segmentation. ViTMAE [56] proposed by He et al. is a scalable self-supervised learner for computer vision. It learned the general presentation of images by masking 75% of the image patches and reconstructing the missing pixels. This study covered only Gram-stain image classification. Although we examined several VT models on two Gram-stain datasets, it might not be enough to draw generalizable conclusions about the effectiveness of visual transformers. In fact, numerous research endeavors have uncovered successful VT applications in the medical domain. Shamshad et al. [57] conducted a comprehensive survey paper recently that summarized studies utilizing VT in the medical domain and over 400 studies were classified based on the problems (e.g., classification, segmentation, registration, etc.) and further categorized by specific tasks (e.g., COVID-19 diagnosis, multi-organ segmentation, etc.) Some researchers have devoted efforts to constructing a novel network architecture or concatenating multiple machine learning models, however, the majority of studies utilize pre-trained transformers models and replace the decision layer for their custom task without modifying the network morphology. The explosive number of publications indicates that VT has permeated every sector of the medical domain and this suggests great potential to develop innovative medical applications. For instance, image-to-text converters have great potential in the medical domain. Tanwani et al. from Google [58] proposed RepsNet which generates automated medical reports in natural language from medical images. Regardless of the advanced architectures of novel deep learning models such as VTs, simple statistical methods or shallow ML algorithms often outperform these models or offer at least a sufficient enough performance, especially on limited, medical domain-specific tasks as demonstrated for anomaly detection in neuroimaging [59] and cross-lingual radiological report classification [60]. Deploying lightweight DL models to an augmented reality (AR) device [61] also has promising applications. For instance, doctors could wear an AR device during surgery to obtain augmented information on a patient, or they can be utilized for training purposes by taking some guidance from the AR device. Lee et al. [62] proposed a transformers-based model that classified one of the three body movements by harnessing electroencephalogram (EEG) signal data and graphics simulated by a head-mounted device. Their model works in a virtual driving environment, and it is feasible to convert it for the medical field in future studies.

## 6. Conclusions

We encourage using VT models for Gram-stained image classification because they could learn the custom images with fewer epochs compared with CNN. With consideration of the model accuracy, models with ViT backbone are recommended as BEiT, DeiT and ViT were outstanding in this study. With regard to the inference time, DeiT small is recommended as the int8 model was able to process six images per second. Finally, the most compact model was MobileViT small, however, we do not recommend using it because of the low accuracy. We recommend the second most compact model, DeiT small in int8, as the accuracy was not degraded regardless of the number of parameters and quantization schemes. Overall, we recommend the DeiT model when we consider test accuracy, inference time and model size for Gram-stained classification. We also advocate using a dataset with 1k or more images, otherwise deep learning models encounter serious overfitting problems. Regarding quantization, per-tensor quantization showed more stable accuracy performances compared with per-channel quantization. We hope this study provides insight to researchers so that they may save time and computational resources in selecting a VT model and determining an optimal configuration, especially for a time-critical application such as Gram-stained image classification.

## Figures and Tables

**Figure 1 biomedicines-11-01333-f001:**
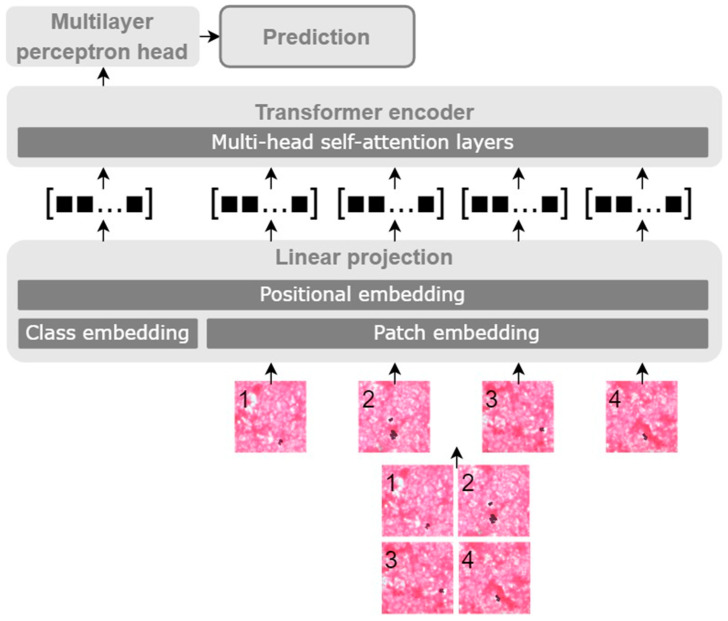
Process of a visual transformer where data flow from the bottom to the top. An input image is split into four patches in this figure for visibility. Each patch is encoded into a predefined size of vectors added by positional vectors and class vectors. The class vector is propagated to the multilayer perceptron head for a decision.

**Figure 2 biomedicines-11-01333-f002:**
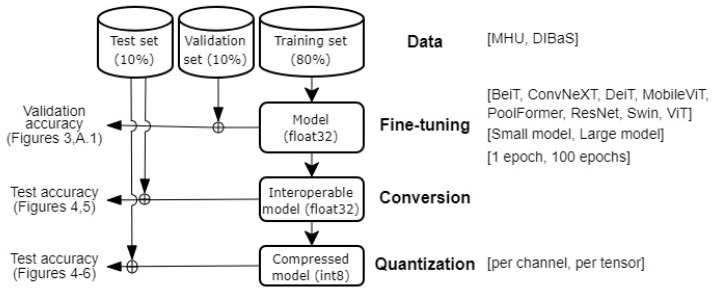
Overview of the study design. Eight models with minimum and maximum parameters were fine-tuned to two custom datasets with two epochs strategies during the fine-tuning phase, while each model was quantized either by channel or tensor during the quantization phase. In total, 128 models were evaluated, which is the Cartesian product of eight model architectures with two parameters on two datasets for two epochs and then two quantization schemes.

**Figure 3 biomedicines-11-01333-f003:**
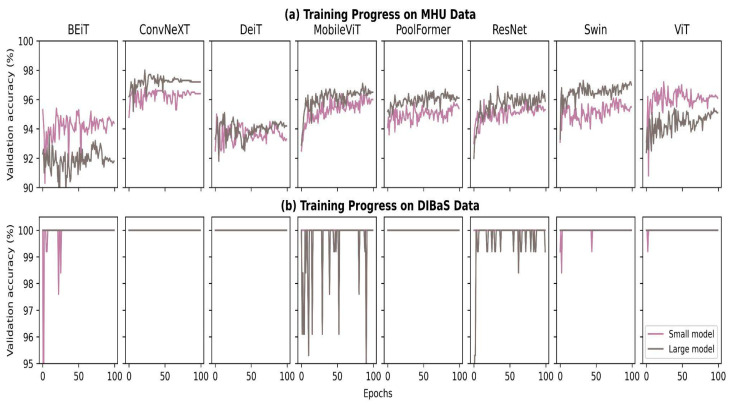
Fine-tuning history for 100 epochs on MHU (**a**) and DIBaS (**b**) datasets. Subplots are organized in correspondence with the alphabetic order of the model name. Parameters for each model architecture are colored either purple (model with minimum parameters) or gray (model with maximum parameters).

**Figure 4 biomedicines-11-01333-f004:**
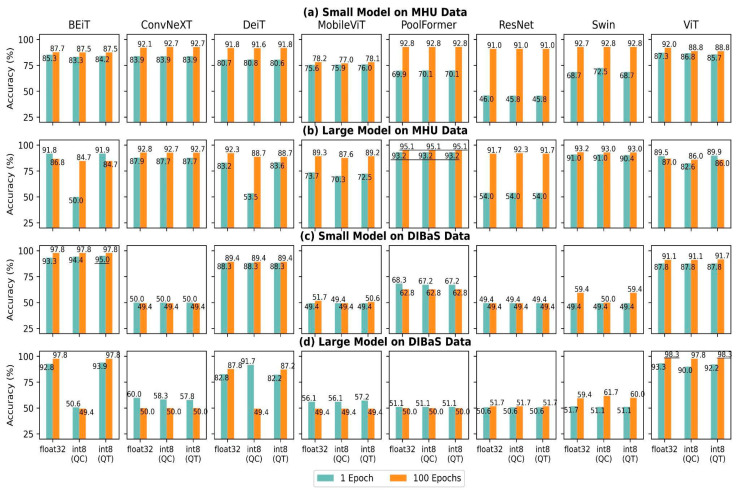
Accuracy of eight models with two parameter setups tuned on MHU and DIBaS datasets. Blue bars indicate the models re-trained for one epoch, whereas orange bars are models re-trained for 100 epochs. Subplots in (**a**,**b**) are the results on the MHU dataset, while (**c**,**d**) are the results on the DIBaS dataset. Models are organized from BEiT to ViT in alphabetic order in the columns with minimum parameters depicted in (**a**,**c**), while those with maximum parameters are shown in (**b**,**d**). Abbreviations: QC, per-channel quantization; QT, per-tensor quantization. Underlining results indicate the overall best models.

**Figure 5 biomedicines-11-01333-f005:**
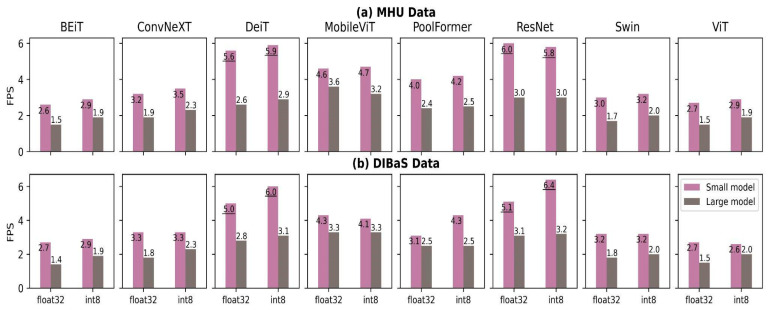
Bar charts of model throughputs on MHU (**a**) and DIBaS (**b**) datasets. Models are color-coded based on the number of parameters (small (purple) vs. large (gray)) and grouped by their bit representation (float32 vs. int8). The y-axis represents the throughput (inference time) measured as the number of processed frames per second (FPS), while int8 indicates per-tensor quantized models. Underlining results indicate high-throughput models, which can process at least five images per second.

**Figure 6 biomedicines-11-01333-f006:**
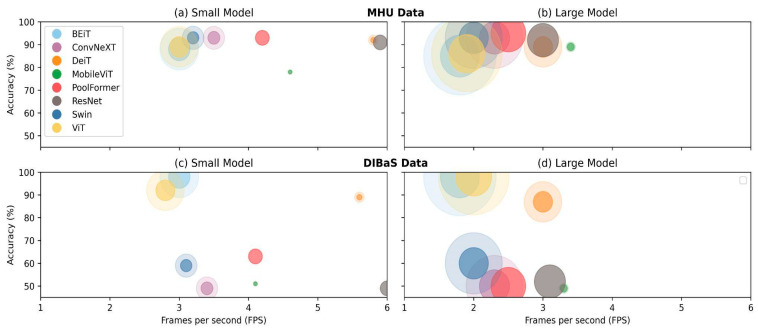
Overview of accuracy, quantization, inference time and model size of eight models with minimum and maximum parameters as bubble charts on MHU dataset (**a**,**b**) and DIBaS dataset (**c**,**d**), respectively. The transparency of colors indicates the model quantization where the semi-transparent color represents the float32 models, while the opaque color represents integer8 models.

**Table 1 biomedicines-11-01333-t001:** Overview of the eight investigated neural network architectures in alphabetical order.

Model	Architecture Traits	Image Size	Patch Size	# Attention Heads	# Parameters (min)	# Parameters (max)
BEiT	Self-supervised VT	224	16	12; 16	86 M	307 M
ConvNeXT	CNN	224	N/A	N/A	29 M	198 M
DeiT	Knowledge distillation VT	224	16	3; 12	5 M	86 M
MobileViT	Hybrid	256	2	4	1.3 M	5.6 M
PoolFormer	Hybrid	224	7, 3, 3, 3	N/A	11.9 M	73.4 M
ResNet	CNN	224	N/A	N/A	11 M	60 M
Swin	Multi-stage VT	224	4	[3,6,12,24]; [4,8,16,32]	29 M	197 M
ViT	Original VT	224	16	12; 16	86 M	307 M

Patch size and attention heads are shown in a single value unless they differ from the parameters. M: million.

## Data Availability

MHU data are publicly available in our repository: https://heibox.uni-heidelberg.de/d/6b672e3ff50a468191b9/ (accessed on 14 April 2023). Codes are available at: https://github.com/kimheekimi/vit_gram_stain (accessed on 14 April 2023). DIBaS data are accessible at: https://github.com/gallardorafael/DIBaS-Dataset (accessed on 14 April 2023).

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
