# Peer review of "Lightweight Visual Transformers Outperform Convolutional Neural Networks for Gram-Stained Image Classification: An Empirical Study"

_biomedicines, 2023, doi:10.3390/biomedicines11051333_

Round 1
Reviewer 1 Report
The paper presents an approach to automate the detection of microorganisms using visual transformers (VT) models, which are a novel architecture in deep learning. However, the paper has several shortcomings that need to be addressed.
Firstly, the paper lacks clarity in its presentation. The introduction fails to provide a comprehensive background of the subject matter and does not sufficiently motivate the research. The research objectives are not well-defined and the methodology is poorly described. The authors need to re-organize the paper to present their ideas in a clear and concise manner.
Secondly, the paper's experimental design is not rigorous enough to draw solid conclusions. The authors only evaluated four visual transformer models and a single convolutional neural network model without any further investigation of other state-of-the-art models. Also, the authors only evaluated their model on a single dataset, which is not enough to draw generalizable conclusions about the effectiveness of visual transformers in Gram stain analysis. The authors need to conduct more comprehensive experiments to evaluate their models.
Thirdly, the authors did not fully analyze the trade-offs between accuracy, inference time, and model size. Although the paper presents some experimental results on the performance of the models, the authors did not provide a thorough analysis of the trade-offs between these factors. This is a critical issue for time-sensitive medical services where quick diagnoses are required, and the authors need to provide more comprehensive analysis in this regard. The authors need to perform statistical analysis to check if the difference between model performances is statistically significant. I recommend to do the non-parametric Friedman test, and then apply the post-hoc Nemenyi test. To visualize model ranks, I suggest to use a Critical Distance Diagram.
In conclusion, while the paper presents a potentially interesting approach to automate Gram stain analysis using visual transformers, the lack of rigor and clarity in the presentation, limited experimental design, and insufficient analysis of trade-offs between accuracy, inference time, and model size make it difficult to draw solid conclusions from the study. The authors need to address these issues to improve the quality and impact of their work.
Author Response
Response to Reviewer 1 Comments
Dear editors and reviewers,
Thank you for your decision letter and suggestions on our manuscript entitled “Lightweight Visual Transformers Outperform Convolutional Neural Networks for Gram-Stained Image Classification: an empirical study” (Manuscript ID: biomedicines-2284460). Your comments and suggestions greatly improved the manuscript. For further details, please refer to the responses below (reviewers’ comments are in bold, responses are in red and changes in the manuscript are in a different color using track changes in Word as required by the journal).
Point 1. Firstly, the paper lacks clarity in its presentation. The introduction fails to provide a comprehensive background of the subject matter and does not sufficiently motivate the research. The research objectives are not well-defined and the methodology is poorly described. The authors need to re-organize the paper to present their ideas in a clear and concise manner.
Response 1: Thank you for your comment. The entire manuscript was revised and it is now in a clear and concise manner. We tried to concise the scope of the manuscript to the emergence of visual transformers models and excluded the deviating topic of interoperability in the introduction. The methodology was also elaborated and the overview of the study design is depicted in Figure 2.
Point 2. Secondly, the paper's experimental design is not rigorous enough to draw solid conclusions. The authors only evaluated four visual transformer models and a single convolutional neural network model without any further investigation of other state-of-the-art models. Also, the authors only evaluated their model on a single dataset, which is not enough to draw generalizable conclusions about the effectiveness of visual transformers in Gram stain analysis. The authors need to conduct more comprehensive experiments to evaluate their models.
Response 2: Now 16 models were evaluated on two datasets in this study. They consist of six VT models and two CNNs with minimum and maximum parameters. We carefully selected those models based on their uniqueness of architecture and the model selection criteria are written in the second paragraph of the study design section and summarized in Table 1.
Point 3. Thirdly, the authors did not fully analyze the trade-offs between accuracy, inference time, and model size. Although the paper presents some experimental results on the performance of the models, the authors did not provide a thorough analysis of the trade-offs between these factors. This is a critical issue for time-sensitive medical services where quick diagnoses are required, and the authors need to provide more comprehensive analysis in this regard. The authors need to perform statistical analysis to check if the difference between model performances is statistically significant. I recommend to do the non-parametric Friedman test, and then apply the post-hoc Nemenyi test. To visualize model ranks, I suggest to use a Critical Distance Diagram.
Response 3: We agree with you that the trade-offs between accuracy, inference time, and model size were not statistically evaluated. However, the non-parametric Friedman test and the post-hoc Nemenyi test were not feasible to apply to our study because the prerequisite of such tests requires at least five measurements (i.e. five datasets) according to the article by Janez Demšar [1]. Besides our local dataset (MHU), we found only one more additional public Gram-stained image dataset (DIBaS). Although we considered creating five test data sets from each dataset, it is statistically still wrong to do so. Janez Demšar [1] stated that the number of test sets should not refer to the test samples drawn from the data set.
[1] Demšar, J. (2006). Statistical comparisons of classifiers over multiple data sets. The Journal of Machine learning research, 7, 1-30.
Thank you for giving us the opportunity to improve our manuscript with your valuable comments. We hope that we addressed all open points.
Sincerely,
Hee E. Kim, MSc
Reviewer 2 Report
This topic is very interesting, but paper needs to be improved. Look at these points:
- "This paper examines four visual transformer models and benchmarks them to one CNN model". and so, what is the purpose of this paper? State it better.
- "The performances of four visual transformer models and one baseline CNN model are reported and examined by accuracy, inference time and model size in this section. All models were re-trained to the Gram-stained images.... subsections." This part seems methods. Move above.
- "Figure 5 (a) could be used as a guideline to choose a reasonable model for Gramstained image classification." Revise this sentence like a regular text sentence.
- Discuss more about the role of "emerging visual transformers" in medicine and surgery. Look at these very important papers on pubmed: -- DOI: 10.1364/BOE.445041 -- doi: 10.3390/ijerph19106347. PMID: 35627884 -- DOI: 10.3390/bioengineering10020186
- "On the other hand, The strength of the Swin model comes from Shifted windows where its acronym was made." what did it lead to? improve this part.
- Conclusion should revise, as it seems a discussion section, rather than a conclusion section. What did the authors find new with their research? report here
Author Response
Response to Reviewer 2 Comments
Dear editors and reviewers,
Thank you for your decision letter and suggestions on our manuscript entitled “Lightweight Visual Transformers Outperform Convolutional Neural Networks for Gram-Stained Image Classification: an empirical study” (Manuscript ID: biomedicines-2284460). Your comments and suggestions greatly improved the manuscript. For further details, please refer to the responses below (reviewers’ comments are in bold, responses are in red and changes in the manuscript are in a different color using track changes in Word as required by the journal).
Point 1. "This paper examines four visual transformer models and benchmarks them to one CNN model". and so, what is the purpose of this paper? State it better.
Response 1: Thank you for your comment. This paper aimed to provide a guideline to researchers and practitioners on VT model selection as well as optimal model configuration for Gram-stained image classification. Now the purpose of the paper is clearly stated in the last paragraph of the introduction.
Point 2. "The performances of four visual transformer models and one baseline CNN model are reported and examined by accuracy, inference time and model size in this section. All models were re-trained to the Gram-stained images.... subsections." This part seems methods. Move above.
Response 2: We agree with you. That paragraph was elaborated and moved to the first paragraph in the study design section (section 3.2).
Point 3. "Figure 5 (a) could be used as a guideline to choose a reasonable model for Gramstained image classification." Revise this sentence like a regular text sentence.
Response 3: The revised sentence is in the fourth paragraph in the discussion. They are now written as follows: Figure 6 is an overview of all results where poor-performance models are placed toward the bottom left corner, while high-performance models are located toward the right top corner. It could provide a gentle guideline for a model selection on a given dataset.
Point 4. Discuss more about the role of "emerging visual transformers" in medicine and surgery. Look at these very important papers on pubmed: -- DOI: 10.1364/BOE.445041 -- doi: 10.3390/ijerph19106347. PMID: 35627884 -- DOI: 10.3390/bioengineering10020186
Response 4: Thank you for your suggestion. The first suggested paper is introduced as a related work in the second paragraph of the introduction. The second and the third recommended papers are mentioned as future studies in the last paragraph of the discussion section. The reference numbers are 9, 60 and 61.
Point 5. "On the other hand, The strength of the Swin model comes from Shifted windows where its acronym was made." what did it lead to? improve this part.
Response 5: Thank you for pointing that out. We noticed that the citation was missing in the initial version of the manuscript. The reference for the Swin model is now cited in reference number 17.
Point 6. Conclusion should revise, as it seems a discussion section, rather than a conclusion section. What did the authors find new with their research?
Response 6: The entire conclusion was revised and the lessons learned from the study are provided there. The essence of our conclusion is as follows: we encourage readers to utilize VT models for Gram-stained image classification because they could learn the custom images with fewer epochs compared to CNN. We also advocate using a dataset with 1k or more images, otherwise deep learning models encounter serious overfitting problems. Regarding quantization, per-tensor quantization showed more stable accuracy performances compared to per-channel quantization.
Thank you for giving us the opportunity to improve our manuscript with your valuable comments. We hope that we addressed all open points.
Sincerely,
Hee E. Kim, MSc
Round 2
Reviewer 1 Report
The authors should formulate recommendations, which Vision Transformer models are better and should be used by researchers for performing similar types of tasks.
Author Response
Author’s response to reviews
Title: Lightweight Visual Transformers Outperform Convolutional Neural Networks for Gram-Stained Image Classification: an empirical study
Manuscript ID: biomedicines-2284460
Dear editors and reviewers,
Thank you for your suggestions on our manuscript entitled “Lightweight Visual Transformers Outperform Convolutional Neural Networks for Gram-Stained Image Classification: an empirical study” (Manuscript ID: biomedicines-2284460). For further details, please refer to the responses below (reviewers’ comments are in bold, responses are in non-bold and changes in the manuscript are in a different color using track changes in Word as required by the journal).
Point 1. The authors should formulate recommendations, which Vision Transformer models are better and should be used by researchers for performing similar types of tasks.
Response 1: Thank you for your comment. We could not perform systematic recommendations because this approach require at least 5 (ideally) separate data sets to be able to infer non-parametric rank-based statistics [1]. Besides our local dataset (MHU), we found only one public Gram-stained image data set (DIBaS) as public Gram-stained data sets are extremely scarce. Nonetheless, the recommendations were elaborated on in a clear manner in the conclusion. The following sentences were added in this revised version: With consideration of the model accuracy, models with ViT backbone are recommended as BEiT, DeiT and ViT were outstanding in this study. With regard to the inference time, DeiT small is recommended as the int8 model was able to process six images per second. Finally, the most compact model was MobileViT small, however, we don't recommend using it because of the low accuracy. We recommend the second most compact model, DeiT small in int8, as the accuracy was not degraded regardless of the number of parameters and quantization schemes. Overall, we recommend DeiT model when we considered test accuracy, inference time and model size for Gram-stained classification.
[1] Demšar, J. (2006). Statistical comparisons of classifiers over multiple data sets. The Journal of Machine learning research, 7, 1-30.
Thank you for giving us the opportunity to improve our manuscript with your valuable comments. We hope that we addressed all open points.
Sincerely,
Hee E. Kim, MSc
Reviewer 2 Report
Authors solved all my criticisms
Author Response
Author’s response to reviews
Title: Lightweight Visual Transformers Outperform Convolutional Neural Networks for Gram-Stained Image Classification: an empirical study
Manuscript ID: biomedicines-2284460
Thank you for your time and prompt feedback. Highly appreciated.
Sincerely,
Hee E. Kim, MSc